# Innate and Adaptive Immunopathogeneses in Viral Hepatitis; Crucial Determinants of Hepatocellular Carcinoma

**DOI:** 10.3390/cancers14051255

**Published:** 2022-02-28

**Authors:** Marco Y. W. Zaki, Ahmed M. Fathi, Samara Samir, Nardeen Eldafashi, Kerolis Y. William, Maiiada Hassan Nazmy, Moustafa Fathy, Upkar S. Gill, Shishir Shetty

**Affiliations:** 1Department of Biochemistry, Faculty of Pharmacy, Minia University, Minia 61732, Egypt; Ahmed.mfathy@mu.edu.eg (A.M.F.); Nardeen.Rafat@mu.edu.eg (N.E.); maiiada_nazmy@mu.edu.eg (M.H.N.); Mostafa_fathe@minia.edu.eg (M.F.); 2National Institute for Health Research Birmingham Liver Biomedical Research Unit and Centre for Liver and Gastrointestinal Research, Institute of Immunology and Immunotherapy, University of Birmingham, Birmingham B15 2TT, UK; 3Department of Biochemistry, Faculty of Pharmacy, Sohag University, Sohag 82524, Egypt; samara.samir@pharm.sohag.edu.eg; 4Department of Internal Medicine, Faculty of Medicine, Cairo University, Cairo 12613, Egypt; kerolos_youssef@cu.edu.eg; 5Barts Liver Centre, Centre for Immunobiology, Barts & The London School of Medicine & Dentistry, QMUL, London E1 2AT, UK; u.gill@qmul.ac.uk

**Keywords:** hepatocellular carcinoma, hepatitis B virus, hepatitis C virus, innate immunity, adaptive immunity

## Abstract

**Simple Summary:**

Infections with Hepatitis B and Hepatitis C viruses are usually asymptomatic and although some patients undergo resolution of infection, the majority do not. Chronic hepatitis leads to continuous cycles of inflammation that can cause complications including liver fibrosis, cirrhosis, and eventually liver cancer. This review summarizes the changes in liver immunity in acute and chronic Hepatitis B and C infections, as well as in liver cancer patients who developed their malignancy within a viral hepatitis background, aiming to better understand the disease biology. This review provides researchers in the field of chronic liver diseases with immune insights into viral hepatitis and hopes to be of help in developing better prophylactic approaches for cancer management.

**Abstract:**

Viral hepatitis B (HBV) and hepatitis C (HCV) infections remain the most common risk factors for the development of hepatocellular carcinoma (HCC), and their heterogeneous distribution influences the global prevalence of this common type of liver cancer. Typical hepatitis infection elicits various immune responses within the liver microenvironment, and viral persistence induces chronic liver inflammation and carcinogenesis. HBV is directly mutagenic but can also cause low-grade liver inflammation characterized by episodes of intermittent high-grade liver inflammation, liver fibrosis, and cirrhosis, which can progress to decompensated liver disease and HCC. Equally, the absence of key innate and adaptive immune responses in chronic HCV infection dampens viral eradication and induces an exhausted and immunosuppressive liver niche that favors HCC development and progression. The objectives of this review are to (i) discuss the epidemiological pattern of HBV and HCV infections, (ii) understand the host immune response to acute and chronic viral hepatitis, and (iii) explore the link between this diseased immune environment and the development and progression of HCC in preclinical models and HCC patients.

## 1. Introduction

Liver cancer is an aggressive tumor and is a leading cause of cancer-related mortality worldwide [1]. The most prevalent type of liver cancer, hepatocellular carcinoma (HCC), accounts for 80–85% of liver cancer cases [2]. HCC usually arises within the background of chronic liver diseases that progress to cirrhosis, which is usually preceded by long-term viral [3] or non-viral [4] liver inflammation (hepatitis). All HCC risk factors are well-defined; however, the number of newly diagnosed HCC cases is on the rise, mainly due to the paucity of effective preventive strategies to limit cancer development in the setting of fibrosis [2]. For many decades, the prevalence of viral hepatitis has been a key factor in driving chronic liver diseases. Chronic viral hepatitis remains a leading cause of HCC development, especially in low- and middle-income countries (LMICs) [5,6]. Hepatitis B virus (HBV) is a hepatotropic virus that causes onset-dependent liver inflammation [7]. Early-onset HBV infection arises in epidemic areas and is usually followed by chronic infection, whereas viral infection contracted during adulthood may progress either to acute illness that resolves or to fulminant liver failure [7]. Infection with hepatitis C virus (HCV), a virus that belongs to the Flaviviridae family, is another known risk factor for advanced liver disease [8,9]. Unlike HBV, most cases of acute HCV infection become chronic with time. Chronic HBV and HCV infections initiate a long-term inflammatory response in the liver, resulting in the activation of active fibrotic changes that can progress to cirrhosis and eventually HCC [8].

Immunotherapy is a concept that was initially introduced in the cancer context in the nineteenth century and was reinforced in the twentieth century by Paul Ehrlich [10]. The breakthrough in immunotherapy in different types of cancers is noteworthy [10]; however, the relevant immunotherapy protocols in HCC remain unclear, partly due to the lack of the complete understanding of the immune landscape in chronic liver diseases and HCC [4]. In this review, we briefly cover the epidemiology, comorbidity, molecular structure, and replication of HBV and HCV viruses. We then provide a comprehensive review of the immune-related pathways in acute and chronic HBV and HCV infections. Finally, we discuss how chronic viral hepatitis alters the liver “immune niche”, predisposing one for HCC development and progression. This review may help to improve the current immune-related treatment strategies used in HCC arising within the background of viral hepatitis and may help us to identify suitable candidates for the currently available HCC management protocols.

## 2. Epidemiology

### 2.1. HBV Epidemiology and Comorbidity

The World Health Organization (WHO) reported that, in 2015, 257 million people (3.5% of the global population) were chronically infected with HBV [11], the majority of whom were born before the HBV vaccine was widely available [11,12]. The highest HBV prevalence is in the Western Pacific and African regions (around 6% of the total number of infected people). This prevalence is lower in the Eastern Mediterranean, Southeast Asia, and European regions, while less than 1% of cases occur in the North and South American regions [11]. With that being said, the implementation of the international HBV vaccine program led to a drastic decrease in the percentage of HBV-infected children; in 2015, just 1.3% of children less than 5 years old developed chronic HBV illness worldwide [11]. The benefits are clear, with a study on the long-term impact of HBV vaccination on more than 300 individuals concluding that the HBV vaccine provides long-lasting immunity and reduces the incidence of HBV-related diseases [13]. However, the high HBV viral load in pregnant mothers, low overall income in LMICs, and inadequate coverage of the HBV vaccine in Africa remain major health challenges [14]. A total of 25% of untreated chronic HBV patients typically die of cirrhosis complications and HCC, a percentage that rises to 50% if only the male gender is considered [15]. In 2015, HBV-related deaths contributed to nearly 61% of the total viral hepatitis-associated mortality [11]. Systematic research for the Global Burden of Disease Study observed that HBV infection directly killed more than 100,000 people of all ages in 2016. That same study demonstrated the high mortality due to complications of chronic HBV infection, where HBV-associated HCC claimed the lives of 349,500 individuals, while HBV-related cirrhosis killed 365,600 patients [16].

### 2.2. HCV Epidemiology and Comorbidity

The global prevalence of HCV infection, as diagnosed by serological testing for anti-HCV antibodies, has been reported to be 1.6% or around 115 million people [17]. However, the percentage of patients with HCV-positive RNA has rapidly declined to about 1% due to advances in novel direct-acting antiviral drugs in viral eradication [18]. The HCV infection rate varies widely across the world, with the highest prevalence being observed in countries with a history of “iatrogenic infections”. Anti-HCV antibodies are detected in more than 5% of adults in Cameroon, Egypt, Gabon, Georgia, Mongolia, Nigeria, and Uzbekistan, where iatrogenic infection is a major risk factor [17]. In Egypt, HCV infection was linked to the intravenous therapy of schistosomiasis (flatworm) 60 years ago [19]. Only a minor proportion of global HCV infections originate in Western countries [17]. HCV has six main genotypes (1–6), with HCV genotype 4 being the most common in low-income countries, while genotype 1 predominates in high- and middle-income countries [17]. The development of an effective HCV vaccine is still the target for many research groups and industrial companies. However, this step is potentially hindered by viral genetic variation, the difficulty of the in vitro culture of the virus, and the lack of appropriate experimental models for testing the effectiveness of newly developed vaccines [20]. Hepatic fibrosis, cirrhosis, and HCC may all be caused by chronic HCV infection. Due to the widespread use of directly acting antivirals (DAAs) and the high sustained virological response (SVR) rates, the incidence of HCC in relation to HCV has reduced, but in those patients with established cirrhosis the absolute risk remains [21].

## 3. Structure, Entry and Replication of Hepatitis B and C Viruses

### 3.1. HBV Mode of Entry and Replication

HBV is an encapsulated DNA virus descending from the Hepadnaviridae family (Figure 1a) [22]. The infection of human hepatocytes with HBV usually starts with the interaction between both viral and host extracellular matrix glycosaminoglycans (GAGs) at the host cell surface. Reversible interaction between the HBsAg high conformational determinant region and the host heparan sulfate proteoglycan (HSPG) initiates the infection process [23]. Directly afterwards, a high-affinity interaction between the viral broad HBsAg (L) protein and the host hepatic bile acid transporter sodium taurocholate co-transporting polypeptide (NTCP) receptor completes the process of entry [24]. The host DNA repair machinery then processes the viral rcDNA genome to covalently closed circular DNA (cccDNA) [25]. Pregenomic RNA (pgRNA), the template for the reverse transcription and translation of core protein and viral polymerases, is exported from the nucleus to assemble with the core protein and polymerase in the cytosol. The rcDNA genome is created and packaged after a multistep process of priming and template shifting inside the nucleocapsid [26]. Loaded with viral genetic material, HBV capsids bind to HBV surface proteins in the host endoplasmic reticulum (ER) before being secreted from the hepatocytes [26]. The production of a large amount of subviral HBsAg particles into the blood, often in the form of enveloped spherical vesicles, is a characteristic feature of HBV replication. Furthermore, during some stages of infection, the translation of the pre-core ORF may result in the presence of a secretory protein (HBeAg) necessary for the settlement of chronicity in the host serum [3].

### 3.2. HCV Mode of Entry and Replication

HCV is a descendant of the Flaviviridae family with a diameter of 45–65 nm [28]. The non-icosahedral nucleocapsid is made up of many copies of HCV core protein and a positive-strand RNA genome of almost 9.6 kb and is surrounded by the envelope (Figure 1b). HCV may bind to low-density lipoproteins (LDL), very-low-density lipoproteins (VLDL), and Apolipoprotein B (APOB) to form lipoviroparticles that initiate host cell entry [29]. Thus far, six main HCV genotypes have been characterized, and each is subdivided into several subtypes and marked by lower-case letters (1a, 1b, etc.) [28]. E1 and E2 glycoproteins and the surface of lipoviroparticles are initially bound to the host’s cell surface LDL receptors and GAGs reversibly. These proteins then form a multi-receptor complex with CD81, SCARB1, claudins, EGFRs, and EPHA2 to initiate cell entry as well as cell–cell transmission [28]. In clear contrast to the hepatitis B viral genome that enters the host nucleus, HCV nucleocapsid is uncoated, releasing positive-strand genomic RNA into the hepatocyte cytosol, where it acts as mRNA for HCV protein synthesis [30]. The translation of the viral RNA ORF provides a precursor polyprotein that has the amino acid sequence for the structural and non-structural HCV proteins [30,31]. Host and viral molecular scissors process the amino acid chain to produce mature viral proteins able to undertake their role in viral replication (Figure 1b) [32]. On the other hand, positive-strand RNA acts as a template to form the intermediate negative-strand, which, in turn, is used as a guide to generate new positive RNA strands that encode polyproteins and form new replication intermediates [33]. Both core and NS5A proteins interact with genomic RNA in cytoplasmic lipid droplets for the de novo synthesis of viral particles. HCV operates the VLDL processing pathway for viral assembly and release [34].

## 4. Host Immune Response to Hepatitis B Infection

### 4.1. Acute Hepatitis B Infection

While vertical HBV transmission mostly leads to chronic infection, infection with the virus later in life tends to resolve before any detectable change or damage to the liver architecture [35]. Within 1 month of acute HBV infection, viral DNA becomes measurable and remains at a low level before HBsAg and HBeAg are serologically detectable. Shortly afterwards, most of the acutely infected individuals clear their viral DNA and their related antigens even before the elevation of the alanine aminotransferase (ALT) levels [36]. Parallel to this, the host develops lifelong immunity against future HBV infections via the development of HBV antigen-specific antibodies [36]. An immature immune system in HBV-infected neonates, HBeAg-induced immune tolerance [37], and viral escape mutations [38] are among the main reasons why HBV infection becomes chronic. Understanding the basic principles of acute versus chronic HBV-mediated activation of the immune system is a milestone in investigating how the HBV-diseased liver microenvironment may trigger malignancy [3].

#### 4.1.1. The Host Innate Immune Response to Acute HBV Infection

Directly after infection, HBV-infected hepatocytes release pathogen-associated molecular patterns (PAMPs) that are recognized by tissue-resident macrophages (Kupffer cells) (Figure 2). This stimulates the nuclear factor kappa light chain enhancer of activated B cells (NF-kB) pathway in the activated macrophages, releasing many proinflammatory proteins, including interleukin-6 (IL6), tumor necrosis factor-alpha (TNFα), interleukin-1 beta (IL-1β), and interleukin-8 (IL-8), in an interferon-independent fashion [39]. IL6 binds to its receptor, IL6R, on the surface of the infected hepatocytes to activate mitogen-activated protein kinase (MAPK), signal-regulated kinase 1,2 (ERK1,2), and c-jun N-terminal kinase (JNK) pathway to inhibit hypoxia-inducible factor-1 alpha (HIF-1α) and HIF-4α transcription factors and subsequently suppress viral replication [39]. Intrahepatic natural killer T cells (NKT) cells produce interferon alpha (IFNα), IFNβ, and IFNγ, which interfere with HBV viral replication [40]. Natural killer (NK) cells also release TNFα, which binds to the FAS receptor on the infected hepatocytes to induce apoptosis. NKT, NK, and macrophages upregulate induced nitric oxide synthase (iNOS) to produce nitric oxide (NO), which also contributes to successful viral eradication [41].

#### 4.1.2. The Host Adaptive Immune Response to Acute HBV Infection

Specific innate immune cells (antigen-presenting cells, APCs) process and present foreign epitopes to naïve CD3 cells that differentiate either into CD8^+^ or CD4^+^ T-cells, initiating adaptive immune response (Figure 2) [42]. This activates different tiers of resistance to viral replication, transcription, or translation to viral proteins. HBV-specific CD8^+^ T-cells release TNFα and IFNγ, which synergistically induce the deamination and destabilization of viral cccDNA in a non-cytolytic cell fashion. This effect is mediated by the paracrine upregulation of the hepatocyte apolipoprotein B mRNA editing enzyme catalytic subunit 3 (APOBEC3) deamination enzyme [43]. Moreover, other cytokines such as tumor necrosis factor-alpha (TNFα), interleukin-6 (IL6), interleukin-1 beta (IL1β), and transforming growth factor beta-1 (TGFβ1) interfere with HBV episomal integrity or with the epigenetic regulation of cccDNA [43,44,45,46]. The therapeutic impact of these cytokines is yet to be justified, partly because of viral episomal DNA resistance to these cytokines [47]. CD8^+^ T-cells also express FAS ligand (FASL), which induces apoptosis in the HBV-infected hepatocytes via the extrinsic FASL/FAS apoptotic pathway. This T cell-mediated clearance of infected hepatocytes results in the serum elevation of ALT 10-15 weeks after infection [48]. Parallel to this, HBV-specific CD4^+^ T-cells promote successful viral clearance by secreting IFNγ and synergizing the function of the CD8^+^ T-cells [49]. Interestingly the early stages of acute HBV are characterized by the induction of the immunosuppressive cytokine IL-10 rather than type-1 IFN, accompanied by a temporary attenuation of NK and T cell responses, which recover once hepatitis B viraemia is reduced [50]. The differentiation of Th_2_ CD4^+^ T-cells that provide B-cell help also confers resistance to acute HBV infection via the secretion of neutralizing antibodies to HBsAg (anti-HBs) [51].

### 4.2. Chronic Hepatitis B Infection

#### 4.2.1. Hepatitis B Resistance to the Host Innate Immunity

To further understand the mechanism of hepatitis B immune evasion, one study focused on a panel of immune genes in biopsies from HBeAg (−) and HBeAg (+) HBV patients, showing the global downregulation of most innate immune genes in chronically infected HBV patients compared to controls [40]. Downregulated genes were related to pattern recognition receptors (PRRs), Toll-like receptors (TLR3), interferon-stimulated genes (ISGs) (GBP1, IFITM1, ISG15), genes involved in type I/III IFN pathway (IFNB1, SOCS1, SOCS3), and chemokines/cytokines (CXCL10, IL6, IL10) [40]. Interestingly, the HBV genotype-based segregation of samples revealed an impaired innate immune response in patients infected with genotypes A and D compared to genotype C carriers [40]. Causative studies showed that HBsAg dampened immune response via the inhibition of TLR9-mediated IFNα production in the human peripheral blood mononuclear cells (PBMCs) of healthy donors (Figure 3) [52]. This effect was mediated by the secretion of TNFα and interleukin-10 (IL10) from peripheral macrophages that inhibited IFNα secretion from dendritic cells [52]. Culturing primary mouse liver parenchymal, endothelial, or Kupffer cells with HBV antigens, hepatitis B virion, or supernatant from HBV-infected cells abrogated TLR3 and TLR4 activation and subsequently inhibited the release of IFN-β and interferon regulatory factor-3 (IRF-3) proinflammatory cytokines in an NF-kB and ERK1/2-dependent manner. Treating HBV-infected cells with small interfering hepatitis B protein target X (HBx) restored inflammatory cytokine production and activated NF-kB and ERK following TLR stimulation [53]. The HBV-mediated immune resistance was not restricted to serum antigens; the terminal protein (TP) domain of HBV polymerase inhibited the IFNα-induced expression of myeloid differentiation primary response gene 88 (MyD88) protein, a signal transducer protein downstream of TLR signaling, in a signal transducer and activator of transcription-1 (STAT1)-dependent manner [54]. The protein levels of TLR2, but not TLR4, were downregulated in liver tissue hepatocytes and Kupffer cells as well as in the peripheral monocytes of HBeAg (+) chronic HBV patients compared to controls. This inhibitory effect was attributed to HBV pre-core or HBeAg proteins that consequently reduced the levels of the antiviral TNFα and the costimulatory molecule CD86 in peripheral blood monocytes and Kupffer cells (KC) [55]. In addition to the role of KCs and TLRs, NK cells also play an important role in HBV. Activated TRAIL-expressing NK cells have been shown to be highly enriched in the liver of patients with CHB, with HBV-infected hepatocytes expressing increased levels of TRAIL-death inducing receptor. These NK cells may therefore contribute to liver inflammation by the TRAIL-mediated death of hepatocytes; this non-antigen specific mechanism is regulated by certain cytokines (e.g., IL-8) during active HBV infection [56]. Counterintuitively, the TRAIL death receptors are also expressed on CD8+ T cells and can lead to rapid NK-mediated elimination; thus, NK cells can negatively regulate antiviral immunity in CHB [57]. A recent study showed that TRAIL can also interfere with HBV replication and expression by dampening the transcription factors HNF4α, PPARα, and RXRα in HepG2 cells transfected with HBV [58]. Other receptor–ligand interactions have also been shown to be important in regulating these innate-adaptive interactions, such as the NKG2D pathway [59]. Type-I interferons may limit this T cell regulation, but this requires further exploration [60,61,62]. Finally, the multipurpose HBx protein prevents viral proteins from being degraded by proteasomes dampening HBV antigen presentation and delaying HBV adaptive immune activation [63].

#### 4.2.2. Hepatitis B Resistance to the Host Adaptive Immunity

Defective T cell function in chronic HBV infection has been attributed to long-term exposure to the antigenic components of the virus and the tolerogenic effect of liver cell compartments [64]. In a mouse model of acute and chronic infection with lymphocytic choriomeningitis (LCMV) virus, chronic antigen exposure led to an altered hierarchy of CD8 T-cell response in non-lymphoid and lymphoid organs. Chronic LCMV antigen exposure and viral load were the main driving force for T-cell malfunction, as evidenced by the disruption of interleukin-2 (IL2) and TNFα production and attenuated T-cell mediated cytopathic effect (Figure 3). While the high level of epitope exposure led to T-cell depletion, lower levels resulted in T-cell exhaustion [65]. Similarly, individuals whose immune system successfully eradicated HBV showed functional mature CD127^high^/programmed cell death-1 (PD1)^low^ CD8^+^ T-cells [66]. In contrast, chronic HBV patients with low viremic levels had a viral-specific PD1^high^ CD8^+^ subpopulation of T-cells that was positively correlated with disease aggressiveness, a condition that was improved after PD1/PDL1 blockade [66]. In a small clinical trial, patients received nivolumab with the HBV therapeutic vaccine GS-4774, which was well tolerated and showed the dampening of HBsAg levels [67]. It is, however, difficult to explain the antiviral role of PD1/PDL1 treatment alone, because most clinical trials including checkpoint inhibitors with other antiviral agents [68]. Likewise, the overexpression of PDL1, the ligand that binds PD1, leading to T-cell exhaustion, on circulating myeloid dendritic cells (mDCs) was observed in chronic HBV patients and strongly correlated with plasma viral load [69]. In vitro blockade of PDL1 signaling upregulated the T-cell inducing factor, IL12, while downregulating the immunosuppressive IL10 cytokine levels [70]. In line with this, it has also been shown that IL-12 can potently augment the capacity of HBV-specific T cells to produce antiviral effector cytokines, and functional recovery mediated via IL-12 is accompanied by the down-modulation of PD-1 [70]. CD4^+^ and CD8^+^ T-cells, amongst other PBMCs from chronic HBV-infected patients, also exhibited a reduced expression of Jumonji domain-containing 6 protein (*JMJD6*) [71]. JMJD6 regulated cell proliferation by decreasing the levels of the cell cycle regression genes cyclin-dependent kinase inhibitor 3 (CDKN3) and small ubiquitin-like modifier 1 (SUMO1) [71]. Longitudinal flow cytometry screening identified a unique T-cell subset that had a higher CD8^+^CD28^−^ and lower CD8^+^CD28^+^/CD8^+^CD28^−^ ratio in chronic HBV patients compared with healthy individuals. This costimulatory CD28 downregulation in HBV-specific cytotoxic T lymphocytes (CTLs) might be associated with chronic exposure to pre-core or HBeAg proteins [72], while HBcAg induces the expression of the immunosuppressive IL10 in human CD4^+^ T-cells [73]. CD25^+^ FOXP3^+^ Treg cells are immunosuppressive T-cells that confer resistance against CTL. An elegant study in an HBV-infected mouse model identified how the innate microenvironment drives Treg function through a distinct F4/80^+^CD206^+^-expressing macrophage population that selectively produces amphiregulin [74]. These amphiregulin-producing macrophages enhance the immunosuppressive effect of Treg cells and abolish the CD8^+^ T-cell-mediated antiviral effects in their experimental settings [74]. Along with overcoming checkpoint inhibitors and the immunosuppressive milieu in the liver environment, several metabolic defects have also been implicated in CHB [75]. Mitochondrial dysfunction has been reported in CHB, where exhausted HBV-specific T cells increase the amount of GLUT-1 receptor on their surface and become dependent on glycolysis for energy supplies, unlike functional T cells, which can utilize oxidative phosphorylation for energy demands. As with the recovery of exhausted HBV-specific T cells with the third signal cytokine, IL-12, this is also able to at least recover mitochondrial function in HBV [76]. HBV-specific T cells have also been seen to undergo metabolic regulation, where the capacity of expanded arginase-expressing myeloid-derived suppressor cells (MDSCs) may impact liver immunopathology in CHB [77]. Recent data have demonstrated a population of tissue resident memory cells (TRM), which are not seen in the circulating compartment, but preferentially expand in patients with a partial immune control of HBV [78]. These tissue-resident populations can be sampled by liver biopsy and fine needle aspirates of the liver, which may provide important information for the longitudinal immune monitoring of novel therapies for HBV and HCC [79]. Lastly, along with T cells, B cells are increasingly being recognized as playing an important part in the control of HBV. The initial knowledge of HBV-specific B cells came primarily from the detection of clinically important serum antibodies. Antibodies to the various protein components of HBV, specifically coat antigens (HBsAg) and nucleocapsid antigens (HBeAg and HBcAg), can be used to diagnose and predict HBV infection. Many B cell functions, including antibody production, antigen presentation, immune regulation, and antiviral activity, are important in the pathogenesis of HBV infection [80]. B cells can produce antibodies such as anti-HBs, anti-HBc, and self-antibodies, which in turn can mediate antiviral immune responses or self-antibodies through various potential mechanisms of CHB infection. AntiHBs antibodies bind to HBsAg and block viral entry and replication. AntiHBs antibody uptakes HBV by binding to HBsAg and inducing the cytophagocytosis of Kupffer cells. Anti-HBs antibody binds to HBsAg and induces perforin/granzyme release from NK cells to eliminate HBV-infected hepatocytes. Anti-HBs antibodies are involved in the formation of immune complexes and bind to dendritic cells, triggering a T cell response. AntiHBc IgG binds to HBcAg and induces hepatocyte lysis via the classical complement pathway. Self-antibodies can also be involved in autoimmune responses, exacerbating liver inflammation [81].

The B cell antigen receptor (BCR) specifically recognizes and binds to HBV antigens, either independently or on the surface of macrophages and dendritic cells. This interaction allows the BCR antigen to be internalized into the endosome. Simultaneously with receptor-mediated endocytosis, MHCI/MHCII molecules produced by B cells converge to form complexes with HBcAg/HBsAg peptides that are processed in the lysosome and delivered to the plasma membrane. Naïve CD4^+^ T cells are activated by B cells presenting the HBcAg MHCII complex to induce a CD4^+^ T cell immune response. In addition, CD8^+^ T cells are activated by B cells presenting the HBcAg/HBsAg MHCI complex to induce a cytotoxic CTL response [81].

However, B cell subsets play a potential immunomodulatory role by inducing or suppressing immune responses through the secretion of various cytokines. B_regs_ can inhibit effector T cell function and enhance Treg cell function by producing IL10. In addition, IL35 secretion by B_regs_ can inhibit the proliferation of naive effector T cells. In contrast, B_eff_ cells produce proinflammatory cytokines, including IL-6, IFNγ, and tumor TNFα, to stimulate memory and memory effector CD4^+^ T cell responses. At the same time, these cytokines may act as non-cytolytic antiviral against infected hepatocytes by inducing cccDNA degradation or reducing HBV transcription. In addition, IL6 secretion by B_eff_ cells may inhibit HBV entry by regulating the expression of an HBV-specific receptor known as NTCP [81].

#### 4.2.3. HBV-HDV Co-Infection Mediated Immunity

In addition to HBV, it is important to consider the impact of hepatitis delta virus (HDV) in relation to HCC development. HDV is the smallest human-infecting virus and characterized as a ‘satellite’ virus dependent on HBV co-infection for persistence. HDV’s RNA genome encodes a protein with two isoforms, which are the small and large delta antigens (S-HDAg and L-HDAg). Eight different genotypes exist for HDV, and although limited research has been undertaken into clinical outcomes on relation to HDV genotypes, some data have suggested that genotype 1 is associated with poorer clinical outcomes including HCC development [82]. HDV utilizes different mechanisms from HBV and HCV mono-infection to modulate the immune system. Similar to HBV, HDV also evades IFN-a mediated immune responses promoting viral persistence and cell survival [83]. HDV antigens on infected cells can be recognized by CD8+ T cells, mediating their killing. The clonal expansion of CD4+ T cells leads to cytokine release (IL-2, IL-10, IFNγ), which stimulates the immune-mediated killing of HDV-infected cells but in an unregulated manner, which also leads to liver necrosis and progressive liver disease [84]. In several human studies, HBV and low-level CD4+ and CD8+ T cell responses have been detected, following in vitro expansion in patients with chronic HDV, and in patients with acute HDV strong CD8+ T cells responses have been detected, indicating the importance of their role in the control of HDV. A recent study analyzing blood and liver samples from HDV-infected subjects revealed a high expression of the innate-like receptor NKG2D on HDV-specific and global intrahepatic CD8+ T cells, associated with TCR-independent activation. NKG2D expression on CD8+ T cells directly correlated with liver inflammation, suggesting a role for non-antigen-specific bystander T cell-related liver inflammation in HDV [85]. Adaptive immune responses overall appear weak in HDV; however, similar to HBV and HCV mono-infection, HBV-HDV co-infected subjects also have a higher frequency of NK cells, with an increased proportion of the immature CD56 bright NK cell population. In line with this, these HDV patients also produce high levels of IFN-γ and TNF-α, which in this regard may be the cause of more aggressive liver disease [86,87]. Unlike HBV and HCV, determinants of the immune response in HDV co-infection are limited, and this remains an area of unmet need for research to find improved treatment strategies and limit HCC development.

## 5. Host Immune Response to Hepatitis C Infection

### 5.1. Acute Hepatitis C Infection

Unlike HBV infection, studies of acute HCV infection in those who work in the healthcare system and studies in experimental Chimpanzee models have demonstrated a high viral replicative capacity during the first days to weeks of infection, a phase that was followed by a plateau stage preceding the development of host immune response [88,89]. Since most acute hepatitis C infections do not resolve, the term “acute HCV infection” describes the first weeks following viral infection and can be further subdivided into early and late acute phases [36].

#### 5.1.1. Host Innate Immune Response to Acute HCV Infection

The early phase of acute HCV infection, dissimilar to HBV, exhibits abnormally high levels of ISGs, with a high viral load probably reflecting unsuccessful viral eradication. While hepatocytes can release IFNs [90], viral PAMPs bind and activate TLR1/2 [91] and RNA helicases retinoic acid-inducible gene-1 (RIG-1)/melanoma differentiation antigen 5 (MDA5) [92] pathways in dendritic cells and macrophages [93]. This, in turn, primes the NF-kB pathway, leading to interferon (mainly type I and III) production and release (Figure 4). Viral resistance to high IFN I and III can be explained by either the NS3-NS5A protease-mediated blockade of the IRF3/IFN I axis or the NS5B-induced inhibition of IFN signaling via the activation of PKR/EIF2α [94]. NK cells also confer important antiviral and immunoregulatory roles in early acute HCV infection [95]. HCV patients bearing compound homozygous genes encoding killer cell immunoglobulin-like receptor; two Ig domains and long cytoplasmic tail 3 (KIR2DL3); and major histocompatibility complex, class I, C (HLA-C1) exhibited a different pattern of viral clearance compared to other genotypes [96]. Regardless of their viral clearance, natural killer group 2 member D (NKG2D^+^) NK cells showed increased IFNγ production and high cytolytic activity in HCV patients compared to control patients (Figure 4) [97]. On the other hand, the activation of NK cells was evident in healthcare professionals who were unintentionally infected with the HCV virus but did not develop acute infection [98]. The viral evasion of NK cell-mediated antiviral activity was initiated by a high HCV E2 protein that crosslinked CD81 on the surface of NK cells, inhibiting their activity [99]. The expression of natural killer group 2, member A (NKG2A), on the surface of NK cells in HCV patients, but not controls, blunted the activation of dendritic cells [100].

#### 5.1.2. Host Adaptive Immune Response to Acute HCV Infection

Unlike HBV infection, the activation of the adaptive immune response to HCV infection takes place several weeks following the infection and is characterized by the activation of different immune components [101,102]. This delayed activation is in contrast to the early activation of the innate immune system components [103]. The infection of chimpanzees with HCV antigens resulted in peripheral and intrahepatic CD4^+^T-cell responses; however, viral clearance was associated only with the intrahepatic CD4^+^T-cell response [102]. Peripheral CD8^+^T-cell responses, on the other hand, were not detectable in any of the experimental animals until after further stimulation with different HCV antigens [102]. Interestingly, both partially and fully HCV viral-eradicating chimpanzees showed the early activation of both peripheral and intrahepatic CD8^+^T-cells producing IFNγ [102]. Although IFN I cytokines were detected in all animals, IFNγ was only detected in animals with complete or partial viral eradication [102]. In another experiment on HCV-infected chimpanzees, the first HCV infection was accompanied with a limited adaptive immune response and the slow onset of viral eradication followed by the development of long-term immunity (Figure 4) [104]. Re-infection to the same animals, however, was quickly eradicated and accompanied by a strong intrahepatic CD8^+^T-mediated cytolytic activity together with an expansion of peripheral CD4^+^ and CD8^+^ memory T-cell response [104]. The indispensable role of intrahepatic CD8^+^T-cells in long-term protection was confirmed by viral persistence following a third dose of HCV infection after CD8^+^T-cell depletion [104]. Moreover, the antibody-mediated depletion of memory CD4^+^T-cells previously resolved HCV infection in chimpanzee prolonged viral clearance after the second reinfection and abrogated CD8+T-cell immunity via the development of viral escape mutations in MHC epitopes [105]. In humans, the expansion of CD4^+^T-cells activates CD38^−^IFNγ^+^CD8^+^T-cells for successful HCV clearance [88]. A close look at the CD4^+^T-cell phenotype associated with viral eradication identified a strong Th1/Th17 response and IL21-expressing CD4^+^Tcell population in HCV-infected individuals [106]. In addition to the T-cell-mediated effect on HCV viral replication, the humoral response may also contribute to acute HCV infection. Human HCV patients always develop neutralizing antibodies against structural and non-structural HCV proteins. This does not, unfortunately, confer any resistance to HCV viral replication [107]. Intriguingly, patients who resolved acute HCV infection showed high levels of these neutralizing antibodies [108] compared to chronic disease carriers, leaving the door open for future studies regarding the role of these antibodies in HCV viral eradication.

### 5.2. Chronic Hepatitis C Infection

#### 5.2.1. Host Innate Immune Response to Chronic Hepatitis C Infection

As in HBV infection, the deficit in the NK cell surface repertoire is associated with HCV viral persistence (Figure 5). For instance, NK cells from chronically infected HCV patients lack NKp46 and NKp30 compared to HBV-infected patients and uninfected controls [109]. This NK phenotype was associated with the low expression of natural cytotoxicity receptor (NCR) and low NK-mediated cytotoxicity compared to patients who cleared HCV infection. NK cells from this cohort also had higher levels of the inhibitory molecule NKG2A, further predisposing them to viral chronicity [109]. STAT1 phosphorylation and activation were evident in NK cells isolated from chronic HCV patients compared to those isolated from control subjects [110]. This IFNα-mediated STAT1 activation in NK cells was accompanied by the expression of suppressor of cytokine signalling-1 (SOCS-1) protein, which negatively regulates IFN signaling and viral eradication [110]. Intriguing results were obtained while studying the NK cells’ malfunction in chronic HCV patients; most, but not all, studies have related this limited anti-viral impact of NK cells to their inability to secrete IFNγ while maintaining their cytolytic mediated impact [111,112]. In vitro studies have shown a low expression of NKG2D, NKp30, and IFNγ on NK cells together with a decrease in their cytolytic activity when directly co-cultured in HCV-infected hepatocytes [113]. Unlike hepatocytes, HCV uptake by macrophages is not CD81-dependent; rather, macrophages engulf HCV particles, leading to the release of the inflammatory cytokines IL6 and IL-1β into the surrounding microenvironment [114]. HCV uptake eventually triggered macrophage apoptosis and death [114]. Cross-talk between macrophages and other cells in the diseased microenvironment can have diverse roles in HCV disease progression. Upon co-culture with HCV-infected hepatocytes, macrophages released CCL5, which activated the expression of several fibrogenic and inflammatory markers in nearby hepatic stellate cells, leading to liver fibrosis [115]. In addition, NK-dependent IFNγ production was induced by pDC-derived IFNα and macrophage-releasing IL15 [116]. In contrast, NS5A-infected apoptotic bodies increased the macrophage-dependent release of IL10 and suppression of IL12 to deactivate NK cells in chronic HCV [117].

#### 5.2.2. Host Adaptive Immune Response to Chronic Hepatitis C Infection

The persistence of HCV infection is linked to an inadequate or early lack of T-cell response, and multi-specific CD4^+^ and CD8^+^ T-cell responses were correlated with HCV viral clearance (Figure 5) [118]. Although patients with chronic HCV infection have abundant virus-specific T-cells, these effector cells are non-functional and associated with viral persistence [119]. HCV core protein may impair T-cell priming by upregulating PDL1 while decreasing TRAIL on the surface of macrophages and dendritic cells [120]. In addition, sub-optimal function and/or a low number of APCs may contribute to T-cell priming failure [120]. In chronic HCV infection, virus-specific CD8^+^ T-cells were characterized by their low proliferation capacity and inability to secrete IFNγ [105,121]. The exhibition of the PD1^high^CD127^low^ CD8^+^ T-cell phenotype was associated with low proliferation and induced apoptosis; however, PD1 check-point inhibition reversed this exhausted T-cell phenotype [122,123]. Blocking CTLA4 and TIM3 co-inhibitory molecules was essential to reactivate strongly inhibited intrahepatic virus-specific CD8^+^T-cells. The distribution of different exhaustion markers on the virus-specific CD8^+^T-cell population in intrahepatic versus peripheral blood of control, HCV chronic patients and disease resolvers were investigated [124]. While PD1 and CD244 were upregulated in the intrahepatic CD8^+^T-cell population in healthy and chronic HCV patients compared to CD8^+^T-cells in peripheral blood, TIM3 co-expression with PD1 and CD244 was characteristic of the intrahepatic CD8+T-cell population in HCV-infected individuals [124]. On the other hand, patients with sustained virological responses downregulate TIM3 and upregulate LAG3 on the surface of their CD8^+^T-cells, showing a proliferative and memory T-cell phenotype [124]. Another study identified the expression of the inhibitory molecules PD1, CD244, KLRG1, and CD160 on the CD127^low^ HCV virus-specific CD8^+^T-cells in half of the chronic HCV patients [125]. Importantly, the CD127^high^ T-cell population showed the downregulation of such inhibitory molecules, emphasizing the need for a T-cell inhibitory cocktail, rather than single-molecule inhibition, to reach a better T-cell activation and viral clearance in chronic HCV patients [125,126]. The role of CD4^+^ T-cells in viral clearance is noteworthy; the number and function of CD4^+^ T-cells were impaired in HCV positive patients compared to HCV patients who resolved the infection [127]. Mechanistic studies detected low levels of IL2 and IFNγ in chronically infected patients compared to controls, further confirming the association between CD4^+^T-cell activation and viral persistence [127]. Another CD4^+^T-cell population, CD4^+^CD25^+^ Treg cells, was abundant in the peripheral blood of chronic HCV patients compared to healthy individuals and HCV resolvers [128]. Mechanistic studies showed that this Treg population functioned in a direct-cell contact pathway to inhibit CD8^+^T-cell activation and functioned independently of IL10 and TGFβ1 [128]. Similar data, yet with a different mechanism of action, were obtained from Cabrera et al., showing an inverse correlation between the number of CD4^+^CD25^+^ Treg cells and the functionality of CD4^+^ and CD8^+^T-cells [129]. Ex vivo and depletion assays, however, showed that these effects were mediated via TGFβ1 and IL10 [129]. The depletion of CD4^+^CD25^+^ Treg cells also enhanced the proliferation and function of the CD8+T-cells isolated from HCV non-resolvers [130]. Interestingly, the virus-specific CCR7^−^CD8^+^ Treg cell population was identified for the first time in chronic HCV patients [131]. This distinct population was antigen-specific and was able to inhibit T-cell response via the IL10 pathway [131]. Although HCV-neutralizing antibodies do exist in the sera of chronically infected patients, HCV exhibited many alternative pathways to bypass this humoral immunity. HCV cell–cell transmission [132], the presence of viral quasi-species [133], the interaction of HCV GAGs with HDL and SCARB1 receptors, and the presence of distinct glycans on the E2 protein may confer resistance against host humoral immunity and drive viral persistence [134,135].

## 6. Viral Hepatitis, Immune Imbalance, and HCC Development

Tumor development is a multistage process during which many host and environmental factors result in malignant clone evolution and proliferation. The “hallmarks” of cancer are a concept initially introduced by Douglas Hanahan and Robert A. Weinberg in 2000 to describe six essential processes acquired during the transformation of normal cells into malignant cells [136]. These steps included the ability of malignant cells to cause independent growth, insensitivity to growth inhibitory signals, replicative immortality, insensitivity to apoptosis, the development of new blood vessels to provide oxygen and nutrients to the newly developed tumors, and the promotion of tumor spread into distant organs [136]. A decade of further investigation of other factors that may also contribute to cancer pathogenesis identified more mechanisms; thereby, Hanahan and Weinberg included both metabolic remodeling and abnormal immune balance to their “hallmark” paradigm, emphasizing the generalized concept of the tumor microenvironment as the “Maestro” of tumor development and progression [137]. HCC is driven by chronic liver diseases that exhibit several rounds of liver inflammation, necrosis, and regeneration making, HCC a paradigm for inflammation-driven cancer [138]. The role of viral [139,140,141] and non-viral [142] inflammation in genetic perturbation and chromosomal aberration predisposing HCC is well-characterized and was comprehensively reviewed in [143]. Instead, the focus of the below section of the review is to summarize the impact of viral hepatitis-mediated immune deregulation on the development and progression of HCC.

### 6.1. The Role of HBV-Related Immunity in the Development of HCC

HBV is a non-cytopathic virus, and the degree of liver damage in chronic HBV infection is driven by the activation of the immune system [144]. The interplay between immune tolerance and immune clearance predicts the progression of chronic HBV disease (Figure 6) [144]. HBV-induced immune deregulation in chronic settings is a fertile niche for the development of HCC.

#### 6.1.1. Innate Immunity and HCC Development

Active mature NK cells have critical antiviral and anti-tumor functional roles [145]; however, an NK cell’s switch to a less active and immature phenotype, confirmed by low cytolytic and low IFNγ production, was detected during HBV chronic infection and HBV-HCC. HBV viral persistence and HBV-induced liver fibrosis upregulated the expression of IL10 and TGF-β immunosuppressive cytokines to dampen the immune surveillance role of NK cells [146], while inducing the expression microRNA-146a in these NK cells [147]. This microRNA-146a induced the further feed-forward inactivation of NK function, as measured by the inhibition of IFNγ and TNFα production in chronic HBV and HBV-HCC patients [147]. In addition, the overexpression of the inhibitory receptors PD-1, TIM3, and NKG2A in NK cells of chronically infected HBV patients predispose them to HCC [148]. Intriguingly, active NK cells may also contribute to hepatocyte damage and HCC development via the IFNγ-dependent activation of epithelial-mesenchymal transition in HB transgenic mice [149], suggesting the importance of immune balance in the fight against HCC [150]. Although NKT cells are not directly associated with HBV-HCC development, they predispose one to malignancy via the induction of liver fibrosis [151]. NKT cells secrete IL4 and IL13 cytokines, activating hepatic liver stellate cells in HBV-transgenic mice [151]. Macrophages play both anti-tumor and tumor roles in HCC development and progression due to their high plasticity and versatile functions [152]. CD137 ligand was found to be upregulated in peripheral monocytes of chronic HBV patients with positive association with cirrhosis [153]. The further investigation of this finding in HBV-transgenic mice revealed the upregulation of CD137 on the surface of the non-specific CD8^+^ memory T-cells, which, in turn, enhanced the recruitment of macrophages to murine livers. Macrophages induced disease progression to liver fibrosis and HCC via the production of TNFα, IL6, and MCP-1 [153].

#### 6.1.2. Adaptive Immunity and HCC Development

The transition of active CD8^+^ T-cells into the exhausted phenotype (characterized by high levels of PD1, CTLA4, and TIM3) is a unique feature of viral persistence and disease progression (Figure 6) [66]. Moreover, in HBV-HCC, the HBV-mediated activation of the STAT3 pathway induced the expression of the oncogene SALL4, which consequently inhibited the expression of microRNA 200c [154]. This microRNA 200c inhibition induced the expression of PDL1, which initiated CD8+ T-cell exhaustion [154]. The high-resolution single-cell sequencing of human HBV-HCC confirmed the presence of two distinct T-cell subtypes within the tumor microenvironment: CD8+ resident memory T cells (T_RM_) and Treg cells [155]. T_RM_ cells expressed high levels of PD1 and were more suppressive and exhausted, while Treg cells expressed the immunosuppressive LAYN protein [155]. HBV patients with the immune-tolerant phenotype showed a high susceptibility to develop HCC compared to immune active HBV patients, confirming the hypothesis that the presence of HBV-specific CD8^+^T-cells that are functionally unable to remove the virus induced continuous inflammation followed by tumor development [156]. In line with this, we have also shown evidence of HBV-DNA integration and clonal hepatocyte expansion in these patients considered immune-tolerant, indicating that hepatocarcinogenesis could be underway even in the early stages of HBV [157]. Moreover, in those subjects deemed as having immune control, evidence of HBV-DNA integration is again present, with the upregulation of genes involved in carcinogenesis [158]. The link between immunity and HBV DNA integration, however, remains tenuous, but with the use of novel technologies (e.g., tissue CyTOF and spatial transcriptomics) we can come to understand more about this relationship in the future. How functionally incompetent CD8 still leads to continuous inflammation is a complex process. Previous studies by Maini et al. have demonstrated that this may be an indirect pathway, where virus-specific CD8 T cells that were unable to control infection led to the recruitment of non-virus specific T cells to the liver, which could drive liver pathology [159]. More recently, further studies have demonstrated that exhausted CD8 cells in human chronic infections and animal models are heterogenous and can be identified by phenotypic and transcriptional markers— e.g., PD1 and TCF1—differentiating them between early and terminally exhausted CD8 T cells [159,160]. The balance of these heterogenous populations may also impact on the associated continuous inflammation. A further animal study demonstrated that the recovery of the immune response may be possible via the specific blockade of the check-point receptor TIGIT and overcoming T cell tolerance to HBsAg, but the consequences are increased liver inflammation and the development of liver tumors [161]. Although the loss of CD4^+^ T-cells in chronic HBV infection was associated with suboptimal HBV-specific CD8^+^ T-cell function and viral persistence [49], the number of circulating and liver-recruited CD4^+^ CTLs increased in the early stages of human HBV-HCC [162]. This T-cell population was reduced and malfunctioned with increasing disease stage, possibly due to the increased infiltration of Treg cells; this loss of CD4^+^ CTLs was associated with poor patient outcomes [162]. This finding supports the notion that the intra-tumoral immune profile is completely different from immunity developed in the corresponding benign liver disease [163,164]. HBV-HCC is usually preceded by liver fibrosis and cirrhosis, which are characterized by the fibrogenic TGFβ pathway signature [165]. The activation of the TGFβ pathway suppressed the expression of microRNA 34a, leading to the upregulation of the Treg-recruiting cytokine CCL22 [165]. Increased Treg recruitment in HBV-HCC patients was associated with poor prognosis and portal vein thrombosis [165]. Treg cells suppressed the function of effector T-cells via the production of immunosuppressive cytokines, including, but not limited to, IL2, 1L10, and TGFβ [166]. Compared to non-viral HCC, the Tregs detected in HBV-HCC uniquely expressed PD-1, which exhausted functional CD8+ T- cells (as marked by low granzyme A/B and perforin and blunted proliferation) within the tumor microenvironment [155,167], explaining the reduced sensitivity of the non-viral HCC to PD1 treatment [168].

CD4+ T-follicular helper cells are a unique anti-tumor immune subset responsible for the activation and maturation of B-cells and B-cell-mediated immunity [169]. The number and functional capacity of the circulating CXCR5^+^CD4^+^ Tfh cells were reduced in patients with HBV-HCC compared to diseased and healthy controls [170,171]. Furthermore, a similar dysregulated pattern was seen relative to the disease stage, as confirmed by low IL21, IL10, and ICOS expression [170,171].

The mechanism of Hepatocarcinogenesis with HDV-HBV co-infection can be mediated via innate and adaptive immune responses as well as epigenetic- and metabolic-related changes. The L-HDAg may facilitate several interactions with signaling pathways involved in survival and apoptosis and promote oxidative stress in the ER. Epigenetic modifications (e.g., histone acetylation and clusterin expression) may aid in HDV-infected cell survival in cancerous cells [172,173].

### 6.2. The Role of HCV-Related Immunity in the Development of HCC

HCV viral persistence promotes immune perturbation in the liver microenvironment through the inhibition of interferon signaling, the skewing of CD4^+^T-cell differentiation towards more hazardous phenotypes, the recruitment of immunosuppressive Treg cells into the liver, and the deactivation of the cytotoxic CD8^+^T-cells. Low-grade chronic inflammation then develops, along with fibrotic and cirrhotic changes with high tumor escape possibilities, altered immune surveillance, and eventually HCC.

#### 6.2.1. Innate Immunity and HCC Development

Following HCV infection, the NF-kB and IFN signaling pathways are activated in NK cells [174]. The first evidence that linked NF-kB pathway activation and HCC development was from a mouse model of chronic liver inflammation in which NF-kB activation in non-parenchymal cells contributed to the tumor burden [175]. Tumor development in this murine model was closely associated with the release of different proinflammatory and protumorigenic cytokines such as TNFα and IL6 [175]. The activation of NF-kB and HCC development is likely to be etiology-independent with our team developing a new dietary-induced HCC experimental model that showed NF-kB activation in the non-tumor tissue of mice developing HCC [176]. IL6 has also been shown to contribute to hepatocarcinogenesis and a study in HCV patients suggested that this may be gender dependent. Unlike male patients, serum IL6 levels in chronic HCV female patients were associated with a high risk of HCC development, linking IL6 levels and gender-related hepatocarcinogenesis [177]. HCV also activated STAT3 pathway in human monocytes/macrophages and dendritic cells in an IL6-dependent manner [178]. Interestingly, a seminal transcriptomic study in HCC patients identified a unique association between NF-kB and STAT3 activation in the adjacent non-tumor tissue, but not the tumor counterparts, and HCC recurrence [179], emphasizing the pivotal role of these pathways in tumor progression and relapse. Additionally, NS5A activates the NF-kB and JNK pathways through TNF receptor-associated factor 2 (TRAF2) [180]. In fact, JNK pathway activation in the non-parenchymal cells was linked with HCC development in experimental mouse models [181]. Furthermore, a large body of evidence associates the activation of the IFN pathway and tumor development [182], whilst lower levels of IL10 cytokine were detected in the sera of HCV-HCC, suggesting that further studies are required to assess whether IL10 supports anti-tumor immunity [183].

#### 6.2.2. Adaptive Immunity and HCC Development

Similar to HBV infection, chronic HCV infection increases the production of lymphotoxin (LT) α and β cytokines and their receptor (LTβR) in hepatocytes. This upregulation led to the increased infiltration of several lymphocytic subsets preceding the development of HCC [184]. A multi-center study of HCV-HCC patients witnessed a high infiltration of CD4^+^T-cells and Treg cells in the fibrous septa and the accumulation of CD8^+^T-cells in the cirrhotic nodules compared to cirrhotic HCV patients. This latter population was associated with decreased levels of NK and NKT cells and with a high risk of tumor recurrence [185]. In contrast, a high prevalence of Treg cells in the peri-tumoral area contributed to an aggressive tumor phenotype [186]. The presence of tumor-associated antigens and tumor-specific mutant antigens is common in HCC [187]. Tumor-associated antigens are recognized and depleted by their specific CD8^+^T-cells. This interaction was associated with longer survival and better patient outcomes [188]. However, constant CD8^+^T-cell exposure to these antigens induced T-cell exhaustion in HCV chronic infection and HCV-HCC [189]. This leaves the door open for the best combination therapy for HCC patients from different etiologies, especially with the lack of significant success of monotherapy with PD1 checkpoint inhibitors in many HCC clinical trials.

## 7. Conclusions and Future Perspectives

The treatment of viral hepatitis has undergone major advances in the last few decades, with effective anti-viral therapy leading to the successful suppression of Hepatitis B replication [190] and remarkable cure rates for Hepatitis C with directly acting anti-virals (DAAs) [191]. Despite this, viral-associated HCC remains a major global problem and many patients continue to develop tumors with active viral infection in many countries and in those who have cured viral infections but established fibrosis/cirrhosis. It is important to further understand the complex immune microenvironment of the liver in the context of viral infection, fibrosis, and carcinogenesis. The underlying etiology of liver disease appears to play a critical role in how patients with HCC respond to therapy, as shown in a sub analysis of Sorafenib trials [192] and highlighted in recent work on NASH-HCC [168,193]. There is also gathering evidence that the success rate of DAAs in HCV eradication falls significantly when the patient has a co-existing active HCC; this could be a consequence of distinct tumor-driven immunosuppression [194]. There are several potential immunotherapy approaches that could have a significant impact in virally induced HCC. For example, widening the repertoire of targeting checkpoint receptor blockade (including TIM-3, NKG2A) and targeting co-stimulatory molecules could augment the recent promising results achieved with PD1-PDL1 blockade. Targeting specific immunosuppressive populations such as Tregs and MDSCs will also be vital in combatting virally induced HCC. We have reviewed the cross-talk between Tregs/MDSCs and CD8 T cells and highlighted factors which could provide promising therapies—i.e., the role of Layilin, as well as the increasing interest in targeting immune-metabolism and how metabolic reprogramming shapes the immune microenvironment of viral hepatitis. In contrast to inhibiting immunosuppressive populations, alternative approaches include driving anti-tumor responses by overcoming NK cell and NKT cell dysfunction.

In this review, we summarized the key immune responses to viral infection in the liver. We now need further understanding of how the development of a tumor changes the immune landscape and how we can use this knowledge to help in stratification/personalized therapy and how to boost the efficacy of immunotherapies such as CAR-T cell therapy and cancer vaccination for patients with HCC. It is also necessary to develop effective preventative strategies for use in patients with virally induced cirrhosis.

## Figures and Tables

**Figure 1 cancers-14-01255-f001:**
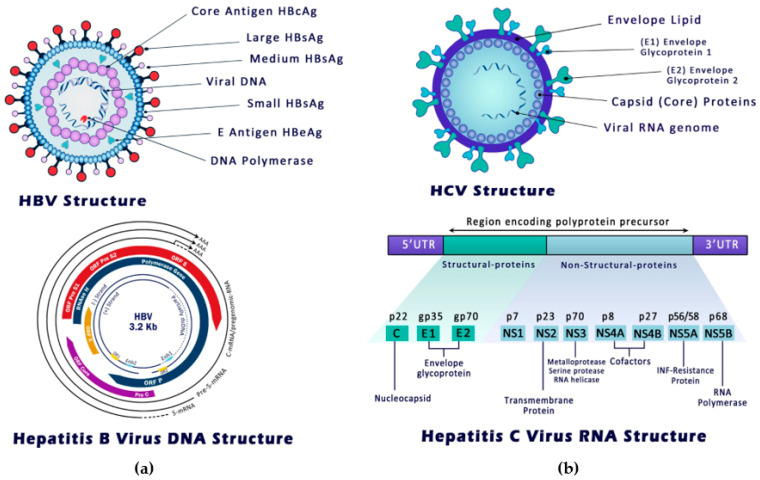
Structure of hepatitis B and hepatitis C viruses. (**a**) Genomic structure of HBV, left panel: The hepatitis B viral envelope encases an icosahedral nucleocapsid, which involves a 3.2-kilobase relaxed circular DNA (rcDNA) genome consisting of a full-length negative strand and partial positive strand. This partially double-stranded HBV genome encodes four open reading frames (ORFs), namely known as P (polymerase), S (surface), C (core), and X (HBx protein) [3]. Genes encoded for HBV genetic materials are PreS1/PreS2/S, which encodes HBsAg; P, which encodes translating polymerase, PreC/C encoding core protein; and X, which encodes HBx protein. Translation of S gene gives large, medium, and small HBsAg, while capsid protein is translated from C gene. HBeAg is translated from the pre-C gene. (**b**) Genomic structure of HCV, right panel: HCV is a 9.6 kb single-stranded positive RNA with one ORF flanked by 2 untranslated regions. The N-terminal ORF encodes HCV structural glycoproteins (core, E1 and E2), whereas the rest of the viral ORF encodes other non-structural proteins (p7, NS2, NS3, NS4A, NS4B, NS5A, and NS5B) [27].

**Figure 2 cancers-14-01255-f002:**
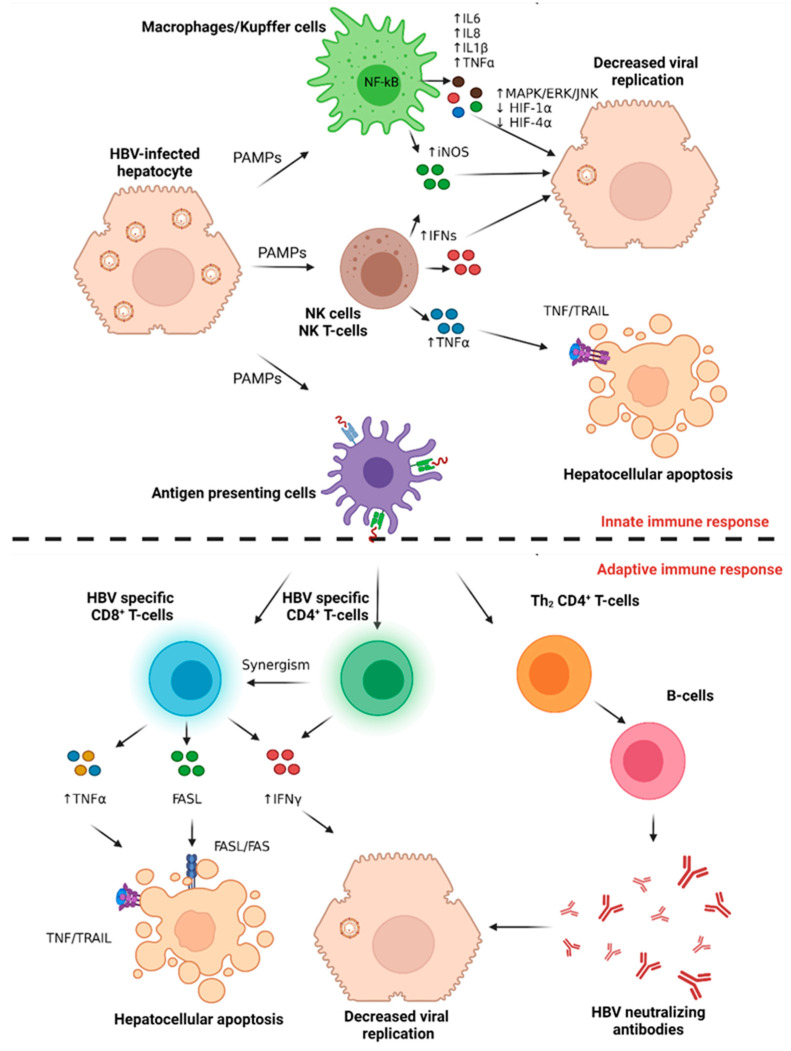
Immune response to acute HBV infection. Various viral antigens activate different adaptive immune cell subsets, initiating the host response to viral entry. PAMPs released from infected hepatocytes activate macrophages/Kupffer cells, NK, and NKT cells to secrete pro-inflammatory and anti-viral cytokines that decrease viral replication while promoting apoptosis in infected hepatocytes. PAMPs also activate antigen-presenting cells to initiate the adaptive immune response and successful viral clearance. Abbreviation: HBV; hepatitis B virus, IL6; interleukin 6, IL8; interleukin 8, IL1β; Interleukin 1, β, iNOS; inducible nitric oxide synthase, TNFα; tumor necrosis factor α, PAMPs; pathogen-associated molecular pattern, IFN; interferons; TRAIL; TNF-related apoptosis-inducing ligand. Figure created with BioRender.com.

**Figure 3 cancers-14-01255-f003:**
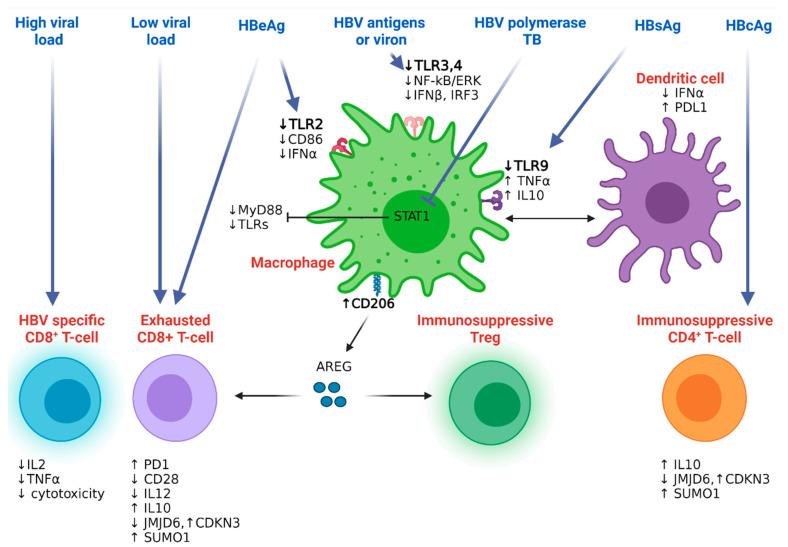
Different immune subsets in chronic HBV infection. TLR signaling molecules are impaired in macrophages exposed to different HBV antigens, leading to the inactivation of dendritic cells and the recruitment of the immunosuppressive Treg cells to the diseased liver microenvironment. High and low viraemia activate low cytolytic HBV-specific CD8^+^T-cells and exhausted CD8^+^T cells, respectively, leading to viral persistence. HBcAg induces the accumulation of the immunosuppressive CD4^+^T-cells that fail to activate the CD8^+^T-cell response. Abbreviations: TLR: Toll-Like Receptor; IFNα: Interferon α; IRF: Interferon Regulatory Factor 3; PD1: Programmed Death-1; PDL1: Programmed Death Ligand 1; IL: Interleukin; JMJD6: Jumonji Domain Containing 6; CDKN3: Cyclin-Dependent Kinase Inhibitor 3; SUMO1: Small Ubiquitin-Like Modifier 1; AREG: Amphiregulin; MyD88: Myeloid Differentiation Primary Response 88. Figure created with BioRender.com.

**Figure 4 cancers-14-01255-f004:**
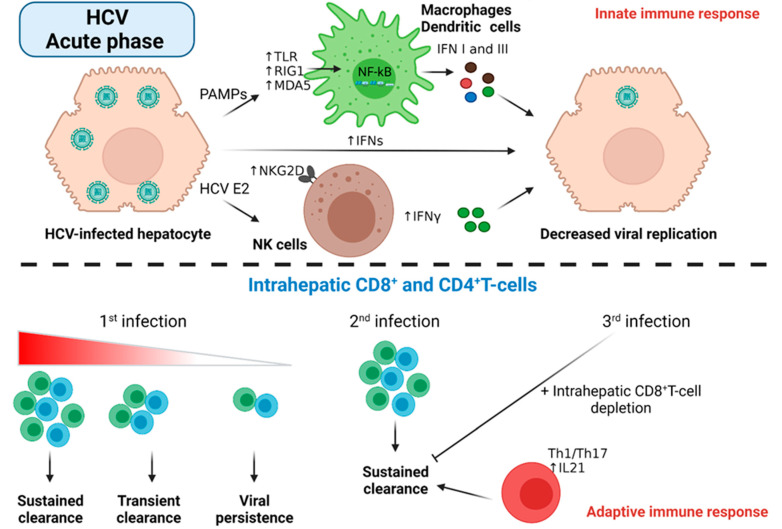
Immune response to the acute HCV phase. Infected hepatocytes release PAMPs, viral antigens, or interferons to activate various innate immune responses. While PAMPs activate interferon signaling in macrophages and dendritic cells, HCV E2 protein activates NK cells to release more IFNs that contribute to the reduced viral replications. The strength of the adaptive immune response to the initial phase of HCV infection detects the infection outcome; a strong intrahepatic T-cell response was associated with viral eradication, while the absence of this adaptive response favors viral persistence. The 2nd and 3rd HCV infections were associated with strong viral eradication and the production of memory T-cell phenotype. Abbreviations: IFN, interferon; TLR, Toll-Like Receptor; RIG1, RNA helicases retinoic acid-inducible gene-1; MDA5, melanoma differentiation antigen 5; IL21, Interleukin 21. Figure created with BioRender.com.

**Figure 5 cancers-14-01255-f005:**
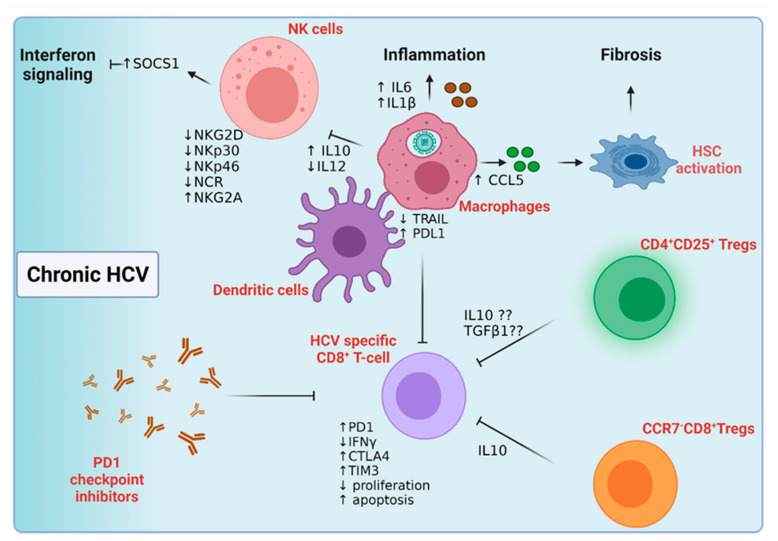
Chronic HCV infection and activation of different immune subsets. Macrophages engulf HCV, releasing CCL5 that activates quiescent hepatic stellate cells to induce liver fibrosis. Macrophages also release the proinflammatory IL6 and IL1β cytokines to the diseased liver microenvironment. An increase in IL10 also diminishes the NK-mediated release of IFNs, leading to viral persistence. HCV-specific CD8+T-cells also acquire an exhausted phenotype due to the abundance of different regulatory T-cell phenotypes and the upregulation of PDL1 on the surface of macrophages and dendritic cells. Abbreviations: SOCS-1, Suppressor Of Cytokine Signaling 1; NCR, Natural Cytotoxicity Triggering Receptor 1; C-C Motif Chemokine Ligand 5; TGFβ, Transforming Growth Factor Beta 1; CTLA4, Cytotoxic T-Lymphocyte Associated Protein 4. Figure created with BioRender.com.

**Figure 6 cancers-14-01255-f006:**
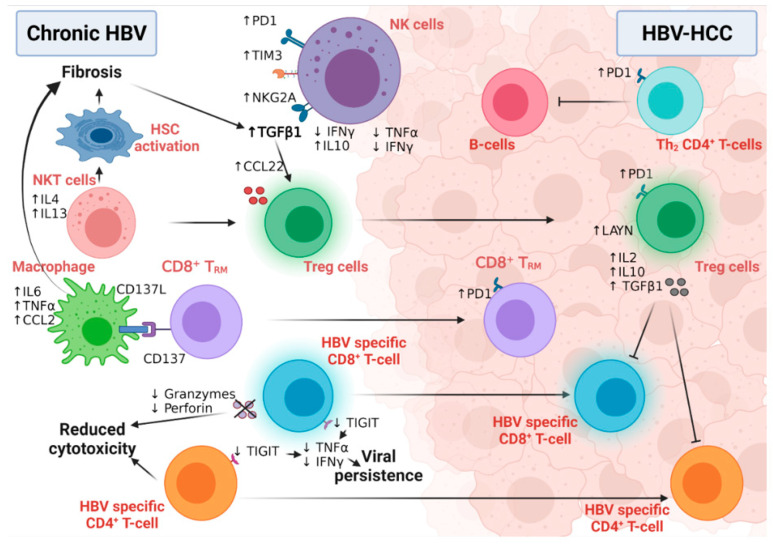
The interplay between different immune cells in HBV-HCC. NK cells in chronic HBV infection and HBV-HCC express different inhibitory receptors and secrete many fibrogenic and proinflammatory cytokines that participate in disease progression. While NKT cells stimulate hepatic stellate cell activation in chronic HBV patients, macrophages also induce fibrosis via interaction with CD8+TRM. HBV-specific CD8+T and CD4+T-cells lose their cytolytic activity and eradicate neither HBV infection nor the HBV-HCC tumor cells. Moreover, Tregs induce their immunotolerant role in both chronic infection and HCC, leading to disease progression and aggressiveness. Figure created with BioRender.com.

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
