# Peer review of "Innate and Adaptive Immunopathogeneses in Viral Hepatitis; Crucial Determinants of Hepatocellular Carcinoma"

_cancers, 2022, doi:10.3390/cancers14051255_

Round 1

Reviewer 1 Report

The review entitled "Innate and adaptive immunopathogeneses in viral hepatitis; crucial determinants of hepatocellular carcinoma" by Zaki, MYW et al. is comprehensively written in the epidemiology and pathogenesis of HBV and HCV, and the development and progression of HCC from HBV and HCV from the aspect of the immunopathogenesis.

One minor comment is that the title of journal is not listed in Reference 119.

Reviewer 2 Report

The aim of this manuscript is to summarize and analyze the epidemiological pattern of HBV and HCV infections, describing the host immune response to acute and chronic viral hepatitis. In this context, this review provides new immune insights in the field of viral hepatitis, with significant future perspectives for the development of preventive and prophylactic therapies.

Even if the manuscript provides an organic overview, with a densely organized structure and based on well-synthetized evidence, there are aspects to be mentioned, to make the article fully readable. For these reasons, the manuscript requires minor changes.

Please find below an enumerated list of comments on my review of the manuscript:

INTRODUCTION:

LINE 80: Hepatitis B Virus is the major cause of acute and chronic liver disease and a global health problem: to this aim, a pivotal role in the contrast of HBV infection is reserved to HBV vaccine, whose main task is to provide a persistent and long-term immunogenicity. In this context, an HBV vaccine not only prevents virus trasmission, but mainly reduces the global burden of HBV – associated disease, as reported by several and recent studies (see, for reference: Mastrodomenico, M.; Muselli, M.; Provvidenti, L.; Scatigna, M.; Bianchi, S.; Fabiani, L. Long-term immune protection against HBV: Associated factors and determinants. Hum. Vaccines Immunother. 2021).

LINE 109: Also for HCV infection, the development of an effective and preventive vaccine against Hepatitis C Virus could have a significant effect on HCV incidence, providing a major contribute to HCV control, as highlighted by different studies (see, for reference: Bailey JR, Barnes E, Cox AL. Approaches, Progress, and Challenges to Hepatitis C Vaccine Development. Gastroenterology. 2019 Jan;156(2):418-430. doi: 10.1053/j.gastro.2018.08.060. Epub 2018 Sep 27. PMID: 30268785; PMCID: PMC6340767).

In conclusion, this manuscript is densely presented and well organized, based on well-synthetized evidences. The authors were lucid in their style of writing, making it easy to read and understand the message, portrayed in the manuscript. Besides, the methodology design was rigorous and appropriately implemented within the study. However, many of the topics are very concisely covered. This manuscript provided a comprehensive review of current knowledge in this field. Moreover, this research have futuristic importance and could be potential for future research. However, I have minor comments only for the introductive section, for improvement before acceptance for publication. The article is accurate and provides relevant information on the topic and I suggest minor changes to be made in order to maximize its scientific impact. I would accept this manuscript, if the comments are addressed properly.

Reviewer 3 Report

This is a very long (if exhaustive) paper dealing with virology and immunology in the liver, as well as to some extent, with HCC.

General points
Section 2 and section 3 are unrelated to the title (and I presume the scope) of the review.
Such sections should be shortened as they do not provide any information pertinent to the topic chosen by the authors. Overall, the manuscript should be clarified with respect to its scope: are immunological processes depicted herein to be focused on HCC or not? Indeed Section 6 that somewhat deals with HCC starts at line 574 and is too long before getting to the point. To my opinion, section 6.1 and 6.2 provide a nice conceptual framework to the phenomenons decribed in this very long review and should be expanded and presented at the inception of the document. Innate immunity and adaptative immunity sections should be inverted since innate immunity is first occurence in immune processes.

Many conceptual shortcuts render many sentences irrelavant to the notions to be conveyed. Distinction between causality and association may have been overlooked in several instances. Many sentences are not readily understandable. Writing quality decreases from the beginning to the end of the paper, where issues are near constantly encountered.

Specific points
Line 63, the causal link between  the fact that HCV infection turns into a chronic disease on the one hand and the fact that HBV and HCV initiate a long term inflammation in the liver is not justified. ‘Therefore’ should be removed and another transition word should be proposed.

Line 95. IFNalpha treatment has been known as obsolete for mayn years in HBV infection and should not be discussed anymore. Please remove. Instead, treatments based on nucleosid analogs should be cited here.

Line 103. The mortality rate due to acute HBV infection is an interesting information that should be further introduced in the paper, since the cause of liver failure and death is most likely immunogenic innthis acute context. However, this statement is in contradiction with line 206 contents (infection with the 206 virus later during life tends to resolve before any detectable change or damage to the liver 207 architecture).

Line 213: The following sentence 'The integration of innate and adap-213 tive immune responses trigger HBV clearance and the formation of effective immunity' is vague and does not convey any effective information.

Line 218:  malignancy is not the unique outcome that results from chronicity, and not the most common.

Section 4.1.2. The fact that the episomal form of HBV is resistant to most if not all of such cytokines and factors should be stated. The notion of viral replication does not convey the current challenges in HBV research since treatment failure upon arrest (due to rebound) is unrelated to its ability to suppress viral replication.

Line 266 and 377: the fact that B cell response has such an important role in the control of HBV infection should be further developed since this is a counter-intuitive information in virology. However, such notion should be put in the following paragrpah and not this one since it is clearly related to the immunology of chronic HBV.

Lne 277. Impaired immune response : could it be due to confounding factors such as social status, ethnicity etc... Indeed African American patients are known to suffer from less active immunity than Caucasian in the United States.

Line 300. What is the role of TRAIL on the virus itself, independently of cell death, if this can be analyzed? Is it directly antiviral and through/on what mechanism/what viral antigen/DNA/RNA population?

Line 303. The sentence 'Moreover, the interaction of innate and adaptive immune responses in the regulating CHB is underscored' is unclear to me. In general, I would try to show that HBV infection being asynchronous from parts to parts of the liver, the overlap of innate and adaptive immune processes may be at the origin of the immunological failure. The NK cell notion that triggers T CD8+ cells elimination is interesting in  that sense. 

Line 336. The results with respect to the clinic of ref 75 should be further detailed. What are the pros and the cons of PD1/L1 therapy in HBV? What are the future directions to circumvent the issues?

Line 400. What is the very function of NKG2D on TCD8+ cells in this context? The causal link between this phenomenon and bystander T cells in unclear to me.

Line 409. Any data of interest on delta Ag-specific B cells?

Line 423. TLRs that are implicated should be identified. 

Ref. 107. Is it a causal link or a correlative association?

Line 475. The sentence 'In humans, HCV clearance coincided with an initial detection of the CD38+IFNγ-CD8+T-cell response that 476 was only activated to CD38-IFNγ+CD8+T-cells after the expansion of CD4+T-cells' is unclear to me.

Ref 117. In what cell type does occur upregulation of SOCS1?

Line 604. The sentence 'HBV viral persistence and HBV-induced liver fibrosis upregulated the expression of IL10 and TGF-β immunosuppressive cytokines to the immune surveillance role of NK cells' is unclear to me.

Ref 154. Are these data correlative or causal?

Line 616. The sentence 'NKT cells also facilitated the infiltration of the immunosuppressive Treg cells with their known pro-tumorigenic roles' suggests that because NKT cells are tumorigenic, they foster Treg accumulation. This is a coarse assumption that should be corrected.

Line 626. I don't see the link between the exhaustion of CD8+ cells and STAT3 in the next sentence.

Line 638. Why are incompetent CD8+ cells able to trigger continuous inflammation? This is a way more indirect process that is acting here and such shortcuts are misleading.

Line 640+ : what is the link between immunity and integration of HBV DNA.

Line 645: sentence is incorrect. IFNg and TNFa are not prooncogenic. What is the vaccination dealt with here?

Line 658. What is the relevance of portal thrombosis herein?

Line 664. What is the 'former group'? Why is this group less sensitive to PD1 treatment?

Line 708/ Ref 185. What is the cell of origin of STAT3/NF-kB?

Line 713/ Ref 189. IL10 levels do not confer protection against HCC. The authors are merely dealing with an association between such levels and the disease onset.

Line 724. The contents of this sentence are tautological.

Section 7 contents is rather general. More insight should be given on pathways, specific targets and clinical tools that could be instrumental in the future.

Round 2

Reviewer 3 Report

The paper is now compatible with publication in the Journal.